# The Lipophilic Purine Nucleoside—Tdp1 Inhibitor—Enhances DNA Damage Induced by Topotecan In Vitro and Potentiates the Antitumor Effect of Topotecan In Vivo

**DOI:** 10.3390/molecules28010323

**Published:** 2022-12-31

**Authors:** Irina A. Chernyshova, Aleksandra L. Zakharenko, Nikolay N. Kurochkin, Nadezhda S. Dyrkheeva, Tatyana E. Kornienko, Nelly A. Popova, Valeriy P. Nikolin, Ekaterina S. Ilina, Timofey D. Zharkov, Maxim S. Kupryushkin, Vladimir E. Oslovsky, Mikhail S. Drenichev, Olga I. Lavrik

**Affiliations:** 1Institute of Chemical Biology and Fundamental Medicine, Siberian Branch of the Russian Academy of Sciences, 630090 Novosibirsk, Russia; 2Engelhardt Institute of Molecular Biology, Russian Academy of Sciences, 119991 Moscow, Russia; 3Federal Research Centre Institute of Cytology and Genetics, Siberian Branch of the Russian Academy of Sciences, 630090 Novosibirsk, Russia; 4V. Zelman Institute for the Medicine and Psychology, Novosibirsk State University, 630090 Novosibirsk, Russia

**Keywords:** DNA repair, topoisomerase I, tyrosyl-DNA phosphodiesterase 1, Tdp1 inhibitor, inhibiting activity, lipophilic nucleosides, topotecan, synergy

## Abstract

The use of cancer chemotherapy sensitizers is a promising approach to induce the effect of clinically used anticancer treatments. One of the interesting targets is Tyrosyl-DNA Phosphodiesterase 1 (Tdp1), a DNA-repair enzyme, that may prevent the action of clinical Topoisomerase 1 (Top1) inhibitors, such as topotecan (Tpc). Tdp1 eliminates covalent Top1-DNA (Top1c) complexes that appear under the action of topotecan and determines the cytotoxic effect of this drug. We hypothesize that Tdp1 inhibition would sensitize cells towards the effect of Tpc. Herein, we report the synthesis and study of lipophilic derivatives of purine nucleosides that efficiently suppress Tdp1 activity, with IC_50_ values in the 0.3–22.0 μM range. We also showed that this compound class can enhance DNA damage induced by topotecan in vitro by Comet assay on human cell lines HeLa and potentiate the antitumor effect of topotecan in vivo on a mice ascitic Krebs-2 carcinoma model. Thereby, this type of compound may be useful to develop drugs, that sensitize the effect of topotecan and reduce the required dose and, as a result, side effects.

## 1. Introduction

Oncological diseases are one of the most common causes of human death in the world. Nevertheless, despite the active development of new treatment methods (cytoreductive therapy, immune therapy, targeted therapy, etc. [1]), radio and chemotherapy based on DNA damage are still the major ones.

One of the important anti-cancer agents in clinical practice is the quinoline alkaloid of the camptothecin family—topotecan. It is currently used as second-line therapy for cervical, ovarian [2] and small cell lung cancer [3]. Its antitumor activity is manifested in the inhibition of the enzyme topoisomerase 1 (Top1). This is a nuclear enzyme that relaxes the positive supercoiling in the DNA molecule. To do this, Top1 generates a break in one of the DNA strands and forms a covalent bond between the tyrosine 723 (human) and the 3′-end of the broken DNA strand at the catalytic site. This intermediate complex promotes fixation of the 3′-end and allows the broken strand to rotate around the intact one until the DNA supercoiling is removed. At this point, the broken ends of the relaxed DNA converge in space, then the 5′-hydroxyl attacks the 3′-phosphotyrosine bond and, thereby, Top1 ligates DNA. Under normal conditions, Top1 cuts and relegates DNA up to 6000 times per minute [4]. Tpc can stabilize the Top1-DNA covalent complex (Top1cc) formed at the first stage of the Top1 catalytic cycle, which leads to the formation of single- and double-strand DNA breaks. For example, double-strand breaks formation occurs when the stabilized complexes are within 10 base pairs on opposite strands of a DNA duplex [5], or when they form adjacent to an already existing single-strand break on the opposite strand [6] or when such a lesion collides with replication forks [4]. The accumulation of DNA breaks eventually leads to apoptosis and the death of cancer cells [7].

The activity of Tpc may decrease due to the mechanisms of hydrolysis of the Top1-DNA bond existing in the cell, which eliminates Top1cc, necessary for the cytotoxic action of Tpc. There are two main ways to remove Top1cc (Figure 1): the exact cleavage of the tyrosyl-DNA bond by phosphodiesterases or the excision of the strand site covalently bound with Top1 by endonucleases (for example, XPF-ERCC1 [8], Mre11-Rad50-Nibrin /Nbs1 (MRN) [9], SLX4 [10] and CtIP [11]). The first pathway in eukaryotic cells is implemented by the enzyme tyrosyl-DNA phosphodiesterase 1 (Tdp1) [12].

The involvement of this enzyme in the development of tumor resistance to Top1 inhibitors has been confirmed by studies on Tdp1-deficient human cell lines and mice. Tdp1-deficient human cell lines and Tdp1-knockout mice were more sensitive to camptothecin and its derivatives [13,14,15,16,17]. On the contrary, when Tdp1 expression is increased, camptothecin causes less DNA damage [18,19,20,21,22]. Therefore, the addition of Tdp1 inhibitors to chemotherapeutic cocktails is a promising strategy for increasing the sensitivity of tumor cells to Top1 inhibitors, in particular Tpc, and reducing their overall toxicity to the body due to a lower therapeutic dose.

One of the promising areas for drug development is the use of natural and modified nucleosides. It is known that this class of substances demonstrates a sufficiently low intrinsic toxicity [23,24], which is important for the development of drugs for combination therapy. Their advantage is also that they probably are able to easily penetrate cell membranes using a nucleoside transporter system similar to that of various nucleoside-based antiviral (acyclovir, zidovudine, etc.) and anticancer drugs (cytarabine, cladribine, gemcitabine) [25].

Previously we have shown that the disaccharide nucleosides with lipophilic groups inhibit Tdp1 in the submicromolar concentration range [26]. Continuing this research, in this work we studied a series of similar monosaccharide derivatives. Moroever, compounds of this class are reported to inhibit enzymes of the PARP family [24,27], which play an important role in maintaining genome stability and cell survival [28,29]. It is known that inhibition of PARP1 by low molecular weight NAD+ analogs leads to suppression of the DNA nucleotide excision repair (NER), and base excision repair (BER) pathways, and also stabilizes the PARP1-DNA complexes [30,31,32]. This causes the DNA replication forks to stop and subsequently leads to DSB [33]—one of the most severe genotoxic damages. Thus, additional inhibition of PARP-family enzymes may lead to the high toxicity of compounds and undesirable side effects. Therefore, all potential Tdp1 inhibitors were additionally tested for their ability to inhibit PARP1/2.

Thus, the aim of our work was to investigate the lipophilic purine nucleosides as the Tdp1 inhibitors and to research whether the leader compounds can sensitize tumor cells in vitro and in vivo to Tpc. To evaluate the inhibiting effect of aromatic and lipophilic derivatives of purine nucleosides on the purified recombinant human Tdp1 enzyme, we conducted screening and kinetic experiments by the fluorescent method. To study the biological properties of the most active inhibitors, we identified their toxicity and the synergistic effect with the antitumor drug Tpc in vitro and in vivo.

## 2. Results and Discussion

### 2.1. Chemistry

It was shown earlier that purine and pyrimidine nucleoside derivatives bearing several benzoic acid residues manifested a high Tdp1 inhibitory effect and low toxicity [34,35]. In order to further investigate the Tdp1 inhibitory effect and anticancer potential of nucleoside compounds several lipophilic purine derivatives with modifications in the purine ring system were synthesized (Table 1). Nucleosides **5c, d** containing hydrocarbon fragments at position six of adenine were obtained from *N*^6^-acetyl-2′,3′,5′-tri-*O*-acetyladenosine by previously elaborated procedures of alkylation of acetylated adenosine derivatives with alkyl chlorides or alcohols under Mitsunobu conditions [36,37,38]. *N*^6^-(1,1′-biphenyl-4-yl)methyl)adenosine (**5a**) was obtained by analogous alkylation from *N*^6^-acetyl-2′,3′,5′-tri-*O*-acetyladenosine and biphenyl-4-methanol under Mitsunobu conditions (treatment with Ph_3_P and *N*,*N*′-diethylazabicarbohylate (DEAD) in tetrahydrofuran with further acetyl deblocking in basic conditions (see Materials and Methods for more details). To obtain *O*-benzoylated derivatives **6a**-**6d** two different strategies were applied. The first strategy was to obtain 2′,3′,5′-tri-*O*-benzoylinosine **6a** by the treatment of inosine with benzoyl cyanide and further chlorination of **6a** to **6b** by the treatment of the former with thionyl chloride (see the experimental part for more details). Product **6b** was characterized by the absence of hypoxanthine NH-group signalled at 12.41 ppm in the ^1^H-NMR spectrum (Appendix A). To obtain 2,6-disubstituted purine derivatives **6c** and **6d** the second strategy was applied. This consisted of glycosylation of the initial 2-fluoro-6-chloropurine and 2,6-chloropurine by 1-*O*-acetyl-2,3,5-tri-*O*-benzoyl-β-d-ribofuranose in the presence of *N*,*O*-bis-trimethylsilylacetamide(BSA) and trimethylsilyltriflate (TMSOTf) under heating with the preferable formation of purine N9-glycosylated products [39,40,41]. This synthetic strategy proposed earlier by Vorbrüggen and co-workers [42] is extensively used for glycosylation of a variety of purine and pyrimidine heterocyclic bases and in our case allowed obtaining pure products **6c** and **6d** in nearly quantitative yields (see the experimental part for more details). The obtained compounds were characterized by the presence of signals in ^1^H-NMR belonging to hydrocarbon, purine moiety, and three moieties of benzoic acid (Appendix A).

### 2.2. Tdp1 Inhibition

In this work, four natural nucleosides (**1**-**4**) and nine aromatic lipophilic derivatives of purine nucleosides (**5a-f**, **6a-d**) were tested for Tdp1 activity. The used test system was based on the enzyme’s ability to remove Black Hole Quencher 1 (BHQ1) from the 3′ end of a 5′ FAM 16-nt oligonucleotide [43]. As a result, an increase in FAM fluorescence was detected and the obtained kinetic curves were used to calculate the IC_50_ parameter (the concentration of the inhibitor at which the enzyme activity is reduced by half). We have found potent Tdp1 inhibitors with IC_50_ in the range from 0.3 to 7 μM (Table 1). Natural purine nucleosides did not affect the activity of Tdp1, which was expected, because they are constantly present in small amounts in the cells. Compounds with bulky aromatic substituents in the heterocycle practically did not lead to a decrease in Tdp1 activity (compounds **5a**, **5b**, **5d**). Interestingly, the addition of a triphenyl methyl group at the 5′ position of the nucleoside in compound **5f** led to a decrease in IC_50_ by several orders of magnitude compared to compound **5b**, which did not have such a substituent (IC_50_ 0.3 ± 0.06 and >50 μM, respectively).

All lipophilic nucleoside derivatives inhibited Tdp1 in the micromolar concentration range, which is consistent with the data we obtained for similar compounds earlier [34,35]. The inhibition of Tdp1 was favorably affected by the introduction of halogens at the second and sixth positions of the heterocycle, and the 2,6-dichloropurine derivative 6d (IC_50_ 0.82± 0.02 μM) turned out to be one of the most potent compounds.

Since it has previously been shown that nucleoside derivatives can inhibit enzymes of the PARP family [24,27], we also checked whether the studied compounds were able to inhibit the enzymes PARP1 and PARP2. As a result, neither of the compounds in this series noticeably reduced the activity of these enzymes up to a concentration of 1 mM (Appendix A). Thus, when using our compounds, one cannot expect side effects from PARP1/2.

### 2.3. The Effect of Tdp1 Inhibitors on Cells’ Viability

Tdp1 inhibitors are intended to be used in therapeutic cocktails to sensitize tumors to the action of anticancer Top1 poisons. Therefore, it is important that they do not display their own toxicity and do not enhance the existing side effects of cancer therapy. To assess the toxicity of the compounds, their effect on the survival of the HeLa cell line (human cervical cancer) was studied using the colorimetric test of metabolic activity of cells EZ4U, which reflects the survival of cells in the presence of the studied compounds. The EZ4U assay is based on the ability of living cells to reduce the colorless tetrazolium salt XTT [sodium 3,3′-{1-(phenylamino)-carbonyl-3,4-tetrazolium) -bis(4-methoxy-6-nitro)-benzene sulfonic acid hydrate] to an orange water-soluble formazan product [44]. This method was used to investigate the intrinsic cytotoxicity of promising inhibitors discovered in the course of this work, as well as the toxicity of their combination with Tpc.

Nucleoside derivatives had moderate or low cytotoxicity against HeLa cells, CC_50_ (50% cytotoxicity concentration) values varied in the range above 30 µM (Table 1). Some compounds had no effect on cell survival at concentrations up to 100 μM.

As already mentioned, we expected to increase the efficacy and/or reduce the dosage of Tpc with Tdp1 inhibitors. To do this, we studied the effect of nucleoside derivatives on Tpc cytotoxicity. We found that at non-toxic or low toxic concentrations (10 μM), the compounds were able to enhance the cytotoxicity of Tpc, but this effect was independent on Tdp1 inhibition ability. Thus, compounds **5a** and **5d** did not inhibit Tdp1, but enhanced the cytotoxic effect of Tpc on cells by two and a half and three times, respectively (Table 1, the last column). We have already observed that compounds that did not inhibit Tdp1 sensitized cells to the action of Tpc [45]. We attribute this to the probable presence of protein targets other than Tdp1 in the cell.

Compound **5f**, on the contrary, was the most potent inhibitor of Tdp1 among the studied compounds, but its sensitizing effect was weakly expressed (1.7 times). The most effective sensitizers were halogen-substituted derivatives containing halogen atoms in the second and sixth positions of the nitrogenous base **6c** and **6d**.

Thus, we took the three most potent Tdp1 inhibitors **5f**, **6c** and **6d** for further investigation. Compound **5f** had the lowest IC_50_ value (0.3 ± 0.06 μM, Table 1). Compounds **6c** and **6d** were also good Tdp1 inhibitors and were the best at sensitizing cells to the action of Tpc. Compound **6b** had an IC_50_ value (2.0 ± 0.6 μM, Table 1) close to three the most active Tdp1 inhibitors but was not a good sensitizer (Table 1). Thus, we did not take it for further experiments.

### 2.4. Type of Inhibition of Tdp1 Enzyme Reaction for the Most Potent Compounds

The first step in understanding the mechanism of action of the leader compounds is to determine their type of inhibition. There are four main types: competitive, non-competitive, mixed and uncompetitive inhibition. A competitive inhibitor is usually similar in structure to a natural substrate, so there is competition between them for binding to the active site of the enzyme. However, unlike the substrate, the competitive inhibitor does not undergo enzymatic transformation. Thus, competitive inhibitors reduce the interaction of the enzyme to the substrate (increase Michaelis constant values, K_M_) without affecting the maximum reaction rate (V_max_). In the case of a non-competitive type of inhibition, the enzyme activity is suppressed by binding the inhibitor outside the active site. The inhibitor causes such conformational changes that do not allow the enzyme to convert the substrate into a product but do not affect the affinity of the enzyme to the substrate. In this case, K_M_ does not change, and V_max_ decreases. In the uncompetitive type, the inhibitor is not attached to the enzyme, it binds only to the enzyme-substrate complex. In this case, an increase in the concentration of the substrate and therefore an increase in the enzyme-substrate complex enhances the inhibitor binding with it. Thus, uncompetitive inhibition is characterized by a proportional decrease in V_max_ and K_M_. There is also a mixed type of inhibition. It is characterized by the fact that the inhibitor affects the binding site and the catalytic center of the enzyme, whether they overlap or not. In a mixed type of inhibition, the dissociation constant increases, and V_max_ decreases. Mixed-type inhibition can be thought of as a combination of competitive and non-competitive inhibition. In this case, both a double enzyme-inhibitor complex and a triple inhibitor-enzyme-substrate complex are formed, and both complexes are catalytically inactive.

To determine the mechanism of Tdp1 inhibition by the most potent Tdp1 inhibitors **5f**, **6c** and **6d**, we studied the dependence of the reaction rate on the concentration of the substrate under stationary reaction conditions in the presence of these compounds. Then, the values of the V_max_ and K_M_ parameters were calculated from the kinetic curves obtained. The conclusion about the type of inhibition was made based on the dependence type of the calculated parameters on the concentration of the inhibitor. As a result, for compounds **5f** and **6d**, a proportional decrease in V_max_ and K_M_ values was observed with an increase in the inhibitor concentration (Appendix A), which is typical for the uncompetitive type of inhibition. Such inhibitors, supposedly, will be more specific inhibitors of Tdp1, because they will bind to the enzyme only in the presence of the enzyme-substrate complex. However, the compound **6c** has nonlinear dependence; in this case V_max_ and K_M_ exponential decrease, which may indicate a more complex mechanism of Tdp1 inhibition by this compound. For this reason, this compound needs additional study of its inhibition mechanism by other methods.

We believe that the uncompetitive inhibition is generally the most preferable, since the binding of the inhibitor to the enzyme-substrate complex is specific and excludes interaction with other enzymes with a similar structure of the active center. Therefore, uncompetitive inhibitors are usually less toxic than inhibitors with other mechanisms. However, in the case of Tdp1, it should be kept in mind that uncompetitive inhibitors mimic the action of the natural mutation H493R, which is considered to be the cause of the neurodegenerative disease spinocerebellar ataxia with axonal neuropathy (SCAN1) [46]. On the other hand, SCAN1 develops for decades, so the use of uncompetitive Tdp1 inhibitors for a short time may not be dangerous, but this requires careful study. 

### 2.5. Influence of the Most Potent Compounds on the Accumulation of DNA Damage by the Alkaline Comet Assay

In the next step, we performed in vitro experiments on the accumulation of DNA damage and the experiments on the influence of the most potent Tdp1 inhibitors on the action of Tpc in vivo (Section 2.6). Compounds **5f**, **6c** and **6d** are the most effective inhibitors according to the screening results (Table 1). Since we did not obtain the sensitizing effect of Tpc on the tumor with compound **6c** (Section 2.6), we did not perform the Comet assay for this compound. We studied the effect of Tdp1 inhibitors **5f** and **6d** on the accumulation of DNA damage by single-cell gel electrophoresis (Comet assay) [47]. Compound **5f** had no sensitizing effect of Tpc in vivo (Section 2.6), but was the most potent Tdp1 inhibitor (Table 1), thus we took it for the Comet assay. This method has proven to be a simple, fast, and a quite sensitive way to detect DNA damage at the level of a single cell, allowing one to analyze almost any eukaryotic cell culture. Several analysis options allow one to visualize various types of DNA damage: single- and double-stranded breaks and alkaline-labile sites [48]. This allows this method to be widely used in fundamental studies of DNA damage and repair [49,50] and in testing the genotoxicity of new chemical and pharmaceutical drugs in vitro and in vivo [51,52]. To study potential drugs, the alkaline version of the method (pH > 13) proposed by Singh and co-authors [53] is widely used. This method allows the detection of all types of DNA damage with high sensitivity [54]. In cells with a large amount of DNA damage, the structural organization and integrity of DNA are disrupted, which leads to its relaxation and/or fragmentation. As a result, cellular DNA in an electric field gives the observed objects the appearance of a “comet”.

Before studying the effect of Tdp1 inhibitors on the accumulation of DNA damage, it was necessary to find out how long the generation of DNA damage by Tpc and their repair take place in HeLa cells. To do this, cell culture was treated with Tpc at 1, 10, and 100 μM concentrations for 30 min, 1, 2, and 3 h and the level of damage to individual cells was monitored by Comet assay. The images of the treated cells are shown in Appendix A. It was shown that Tpc caused the greatest amount of damage after 30 minutes’ treatment. After this time, the repair systems began to correct these lesions and as a result, after 3 h most of them have been eliminated (Appendix A). According to this result, it was further decided to treat cells with 10 μM of Tpc for an hour. The level of DNA damage at this point was enough to be detected, the data were reproducible, and can be evaluated by the Comet assay. The effect could be enhanced by the tested compounds without leading to cell death. It was important for us to see this enhancement in the presence of our compounds.

Further, we investigated the effect of Tdp1 inhibitors named **5f** and **6d** on the DNA damage amount caused by Tpc. Treatment with compounds **5f** and **6d** in concentrations up to 1 mM did not affect the integrity of DNA (the percentage of DNA in the tail for cells treated only with inhibitors was comparable to the same parameter for untreated cells or cells treated with 1% DMSO, solvent for compounds, Appendix A). When treated with a combination of 10 μM Tpc and **5f**, the amount of DNA damage (% DNA in tail) did not significantly differ from the amount of damage generated only by Tpc (Figure 2). Compound **6d**, in contrast, increased the amount of DNA damage with an increase of the concentration in combination with 10 μM Tpc (Figure 2).

In the further experiments, the concentration of Tpc and compound **6d** was reduced and the degree of DNA damage was monitored (Appendix A). It was shown that the percentage of DNA damage was similar in experiments where cells were treated with only Tpc 10 μM and a combination of Tpc 1 μM and 6d 5 μM (Figure 3). Thus, in vitro, it was possible to reduce the dose of the chemotherapy drug required 10 times to produce a serious level of DNA damage.

Despite the data on the sensitization of the cytotoxic effect of Tpc by Tdp1 inhibitors, the evidence of the influence of the latter on Tdp1 in the cell needs further investigation.

### 2.6. The Influence of Lead Compounds on the Action of Topotecan In Vivo

The effect of compounds **5f**, **6c** and **6d** on the effectiveness of Tpc was studied in vivo on a mice ascitic Krebs-2 carcinoma model. Krebs-2 carcinoma can be transplanted into mice of various genetic constitutions. With intraperitoneal inoculation, it forms an ascites form and does not metastasize [55,56]. This model combines the advantages of in vitro and in vivo approaches to study the cytotoxic effects of compounds, since ascites cells grow in the environment of the body (in vivo), and intraperitoneal administration of the drug ensures its direct contact with tumor cells (in vitro), bypassing absorption, delivery to the tumor, possible transformations, etc.

During the experiment, mice were injected intraperitoneally with 2 × 10^6^ ascites cells, after that a liquid ascites tumor grew at the site of inoculation. Next, the mice were divided into a control group that did not receive treatment, and four groups that received treatment once on the fourth day after tumor inoculation: (1) vehicle (DMSO solvent 15% + Twin-80 10% in saline); (2) Tpc 1 mg/kg of body weight intraperitoneally; (3) Tpc, as described above, and compounds 100 mg/kg intraperitoneally; (4) only compounds as described.

The effect of drugs on the tumor was assessed at the end of the experiment by the weight of the ascitic tumor (the difference in the weight of the mouse with the tumor and after tumor removal), as well as by the number of tumor cells in the ascites. The combined treatment by Tpc and compounds **5f** and **6c** did not result in a reliable decrease in the tumor mass and the number of tumor cells compared to only Tpc (Appendix A). In contrast, the compound **6d** in combination with Tpc led to a significant decrease in the tumor characteristics compared to only Tpc (*p* = 0.0018) (Figure 4), while the Tdp1 inhibitor itself did not affect this parameter.

We have developed Tdp1 inhibitors for sensitizing tumors to known anticancer drugs. It is known that Tdp1 can contribute to the drug resistance in some cancers and the hypothesis that Tdp1 inhibitors can increase the cytotoxicity of Top1 poisons, was confirmed by studies involving Tdp1-knockout mice and human cell lines [13,14,15,16,17]. Previously we identified effective Tdp1 inhibitors from different classes of compounds that exhibited Tdp1 inhibitory activity in the low micromolar and submicromolar range with low intrinsic cytotoxicity. In our review [57] we discussed the ability of this and other different approaches of developing Tdp1 inhibitors as potent prodrugs. We believe that the fact that Tdp1 inhibitors of different chemical types sensitize the antitumor effect of Tpc indicates that our hypothesis is correct. We have previously shown that usnic acid and coumarin derivatives also increase the antitumor activity of topotecan [58,59,60]. This research has revealed a set of lipophilic purine nucleosides as Tdp1 inhibitors. In this work the compound **6d** proved to be an effective Tdp1 inhibitor and a good sensitizer of Tpc both in vitro and in vivo. We plan to continue detailed studies of the action of the compound **6d** as a potential anti-cancer drug used in a cocktail with Top1 poisons.

## 3. Materials and Methods

### 3.1. Chemistry

The solvents and materials were reagent grade and were used without additional purification. Column chromatography was performed on silica gel (Kieselgel 60 Merck, Germany, 0.040–0.063 mm). TLC was performed on an Alugram SIL G/UV254 (Macherey-Nagel, Düren, Germany) with UV visualization. The ^1^H and ^13^C NMR spectra were recorded on a Bruker (Karlsruhe, Germany) AMX 300 and 400 NMR instruments at 298 K. The chemical shifts in ppm were measured relative to the residual solvent signals as internal standards (CDCl_3_, ^1^H: 7.26 ppm, ^13^C: 77.16 ppm; DMSO-*d*_6_, ^1^H: 2.50 ppm, ^13^C: 39.52 ppm). Spin-spin coupling constants (*J*) are given in hertz (*Hz*). The high-resolution mass spectra (HRMS) were registered on a Bruker micrOTOF II instrument using electrospray ionization (ESI). The measurements were taken in a positive ion mode (interface capillary voltage 4500 V); mass range from *m*/*z* 50 to *m*/*z* 3000 Da; internal calibration was undertaken with electrospray calibrant solution (Fluka). A syringe injection was used for solutions in acetonitrile: water mixture, 50/50 vol. % (flow rate 3 mL/min). Melting points were measured on Electrothermals melting point apparatus (Great Britain) and were uncorrected. Nitrogen was applied as a dry gas; interface temperature was set at 180 °C. Nucleosides **1**-**4** and 1-*O*-acetyl-2,3,5-tri-*O*-benzoyl-β-d-ribofuranose (CAS 6974-32-9) were purchased from *Sigma Aldrich* company (https://www.sigmaaldrich.com, accessed on 10 July 2020). *N*^6^-Acetyl-2′,3′,5′-tri-*O*-acetyladenosine (Ac-acetyl group) was obtained according to the literature procedure [61]. 2-Fluoro-6-chloropurine and 2,6-dichloropurine were synthesized according to [62]. 2′,3′,5′-Tri-*O*-benzoylinosine (**6a**) was obtained by benzoylation of initial inosine with benzoyl cyanide acording to procedure, published by Prasad et al. [63]. Trityl-(**5b**, **5f**), antranyl-(**5c**) and pyrenyl-(**5d**) nucleoside derivatives were synthesized in accordance with the earlier described procedures [36,37,38].



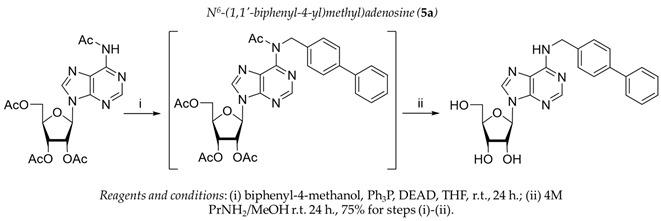



To the solution of *N*^6^-acetyl-2′,3′,5′-tri-*O*-acetyladenosine (300 mg, 0.68 mmol), Ph_3_P (358 mg, 1.36 mmol) and biphenyl-4-methanol (252 mg, 1.37 mmol) in THF (5 mL), DEAD (0.213 mL, 1.36 mmol) was added in one portion and the solution was kept at r.t. for 48 h. The reaction was monitored by TLC (silica gel, CH_2_Cl_2_:EtOH − 97:3 (%)). After 48 h the reaction mixture was evaporated and the residue was dissolved in CH_2_Cl_2_ and washed with brine (2 × 20 mL). The organic layer was separated, dried over anhydrous sodium sulfate, filtered and evaporated in a vacuum. The residue was applied to column chromatography (silica gel, CH_2_Cl_2_:EtOH − 97:3 (%)). The partially purified compound was dissolved in 4M PrNH_2_ in MeOH solution (8.5 mL) and left at r.t. for 24 h, after which the mixture was evaporated and the residue was applied to column chromatography on silica gel. The column was washed with CH_2_Cl_2_:EtOH (95:5 (%)), the product was eluted with CH_2_Cl_2_:EtOH (90:10 (%)) to give **6** as a white powder. Yield for two steps 217 mg (75%). M.p. = 176–178 °C. ^1^H-NMR (300 MHz, DMSO-*d*_6_): 8.46 br s (1H, NH), 8.38 br s (1H, H-2), 8.22 s (1H, H-8), 7.64–7.56 m (4H, H2,H3,H5,H6-biphenyl), 7.48–7.40 m (4H, H8,H9,H11,H12-biphenyl), 7.33 tt (*J*_H10-H11_ = *J*_H10-H9_ = 7.3, *J*_H10-H12_ = *J*_H10-H8_ = 1.2, H10-biphenyl), 5.90 d (1H, J_1′2′_ = 6.1, H-1′), 5.41 d (1H, *J*_2′-OH_ = 6.2, 2′-OH), 5.34 dd (1H, *J*_5′a-OH_ = 4.7, *J*_5′b-OH_ = 7.0, 5′-OH), 5.15 d (1H, *J*_3′-OH_ = 4.7, 3′-OH), 4.76 br s (2H, NCH_2_), 4.62 ddd (1H, *J*_2′-OH_ = 6.2, *J*_2′1′_ = 6.1, *J*_2′3′_ = 5.0, H-2′), 4.16 ddd (1H, *J*_3′-OH_ = J_3′2′_ = 5.0, *J*_3′4′_ = 3.4, H-3′), 3.97 ddd (1H, *J*_4′3′_ = *J*_4′5′a_ = *J*_4′5′b_ = 3.4, H-4′),3.68 ddd (1H, *J*_5′a5′b_ = −12.1, *J*_5′a4′_ = 3.4, *J*_5′a-OH_ = 4.7, H-5′a), 3.55 ddd (1H, *J*_5′b5′a_ = −12.1, *J*_5′b4′_ = 3.4, *J*_5′b-OH_ = 7.0, H-5′b). ^13^C-NMR (75 MHz, DMSO-*d*_6_): δ = 154.54 (C6), 152.34 (C2), 148.50 (C4), 139.92 (C8), 139.27, 138.59, 128.85, 127.70, 127.22, 126.53 (biphenyl), 119.78 (C5), 87.94 (C1′),85.88 (C4′), 73.48 (C2′), 70.62 (C3′), 61.64 (C5′), 42.62 (NCH_2_). HRMS: *m*/*z* [C_23_H_23_N_5_O_4_+H]^+^ calculated 434.1823, found 434.1820.



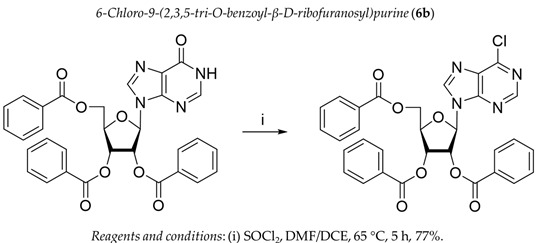



For synthesis of compound **6b** a mixture of dry DMF (1.7 mL, 24.5 mmol), thionyl chloride (3.3 mL, 16.9 mmol) and 2′,3′,5′-tri-*O*-benzoilinosine (2.8 g, 4.8 mmol) in 52 mL of 1,2-dichloroethane (DCE) was heated at 65 °C in an oil bath for 5 h. The reaction mixture was then cooled to ambient temperature, diluted with ethyl acetate and washed successively with a saturated solution of sodium bicarbonate (50 mL) and cold water (2x50 mL), dried over sodium sulfate, filtered off and evaporated in a vacuum to dryness. The residue (cinnamon foam) was applied to chromatography on silica gel in dichloromethane/dichloromethane-ethanol 100–95:5 (*v*:*v*) to obtain 2.2 g (77%) of 6-Chloro-2′,3′,5′-tri-*O*-benzoyl-β-d-ribofuranosylpurine riboside as a foam. ^1^H-NMR (300 MHz, CDCl_3_): δ = 8.61 s (1H, H-2), 8.27 s (1H, H-8), 8.07 dd (2H, ^3^*J* = 8.4, *^4^J* = 1.3, ortho-H-Bz), 8.02 dd (2H, *^3^J* = 8.4, *^4^J* = 1.3, ortho-H-Bz), 7.92 dd (2H, *^3^J* = 8.4, *^4^J* = 1.3, ortho-H-Bz), 7.63–7.50 m (3H, para-H-Bz), 7.47–7.32 m (6H, meta-H-Bz), 6.45 d (1H, *J*_1′2′_ = 5.1, H-1′, H-2′), 6.41 dd (1H, *J*_2′21′_ = 5.1, *J*_2′3′_ = 5.4, H-2′), 6.25 t (1H, *J*_3′2′_ = *J*_3′4′_ = 5.4, H-3′), 4.94 dd (1H, *J*_5a′5′b_ = −12.2, *J*_5′a4′_= 3.3, H-5′a), 4.85 ddd (1H, *J*_4′3′_ = 5.4, *J_4′5′b_* = 4.2, *J*_4′5′a_ = 3.3, H-4′), 4.70 dd (1H, *J*_5′b5′a_ = −12.2, *J*_5′b4′_= 4.2, H-5′b). ^13^C NMR (75 MHz, CDCl_3_): 166.20 (C=O, 5′*O*-Bz), 165.42, 165.25 (C=O, Bz), 152.43 (C-2), 151.70 (C-4), 151.37 (C-6), 144.21 (C-5, C-8), 134.03, 133.93, 133.63, 129.94, 129.82, 128.71, 128.68 (Ph), 87.70 (C-1′), 81.15 (C-4′), 73.98 (C-3′), 71.50 (C-2′), 63.40 (C-5′). HRMS: *m*/*z* [C_31_H_23_ClN_4_O_7_+H^+^] calculated 599.1328, found 599.1326, [C_31_H_23_ClN_4_O_7_+Na^+^] calculated 621.1147, found 621.1144, [C_13_H_23_ClN_4_O_7_+K^+^] calculated 637.0887, found 637.0882.



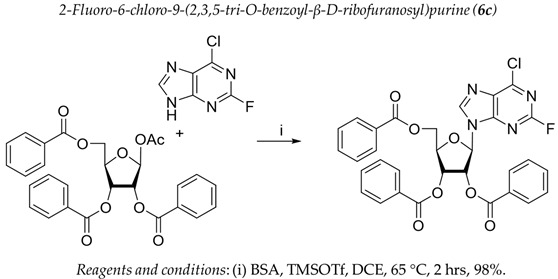



For synthesis of compound **6c** to a suspension of 0.354 g (2.052 mmol) 2-fluoro-6-chloropurine in 20 mL of DCE, 0.950 g (1.883 mmol) of 1-*O*-acetyl-2,3,5-tri-*O*-benzoyl-β-d-ribofuranose and 0.610 mL (2.495 mmol) of *N*,*O*-bis-trimethylsilylacetamide (BSA) were added. The resulting mixture was stirred for approximately 30 min at r.t. until homogenization and trimethylsilyltriflate (TMSOTf, 0.5 mL, 2.756 mmol) was then added the mixture and the mixture was heated at 65 °C in an oil bath for 2 h. The resulting mixture was cooled to r.t., diluted with ethyl acetate, washed successively with a saturated sodium bicarbonate solution (50 mL), water (50 mL), brine (50 mL), dried over sodium sulfate, filtered and evaporated in a vacuum to dryness. The resulting residue (yellowish foam) was purified chromatographically on silica gel in dichloromethane/dichloromethane-ethanol 100 −100:5 (*v*:*v*) to obtain 1.14 g (98%) of 2-Fluoro-6-chloro-9-(2,3,5-tri-*O*-benzoyl-β-d-ribofuranosyl)purine as a foam. ^1^H-NMR (300 MHz, CDCl_3_): δ = 8.25 s (1H, H-8), 8.08 dd (2H, *^3^J* = 8.4, *^4^J* = 1.3, ortho-H-Bz), 8.02 dd (2H, *^3^J* = 8.4, *^4^J* = 1.3, ortho-H-Bz), 7.91 dd (2H, *^3^J* = 8.4, *^4^J* = 1.3, ortho-H-Bz), 7.63–7.53 m (3H, para-H-Bz), 7.49–7.41 m (4H, meta-H-Bz), 7.36 dd (2H, *^3^J*_1_ = 8.4, *^3^J*_1_ = 7.5, meta-H-Bz), 6.43 d (1H, *J*_1′2′_ = 5.5, H-1′), 6.22 dd (1H, *J*_2′3′_ = 5.7, *J*_2′1′ = _5.5, H-2′), 6.13 dd (1H, *J*_3′2′_ = 5.7, *J*_3′4′_= 4.6, H-3′), 4.91 dd (1H, *J*_5a′5′b_ = −12.2, *J*_5′a4′_ = 3.2, H-5′a), 4.85 ddd (1H, *J*_4′5′b_ = 4.0, *J*_4′5′a_ = 3.2, *J*_4′3′_ = 4.6, H-4′), 4.72 dd (1H, *J*_5′b5′a_ = −12.2, *J*_5′b4′_ = 4.0, H-5′b). ^13^C NMR (75 MHz, CDCl_3_): 166.20 (C=O, 5′*O*-Bz), 165.40, 165.26 (C=O, Bz), 157.43 (*d*, ^1^*J*_C-F_ = 221.9, C-2), 153.29 (*d*, ^2^*J*_C-F_ = 16.7, C-6), 144.16 (C-8), 144.09 (*d*, ^2^*J*_C-F_ = 16.4, C-4, overlaps with C-8), 131.07 (d, ^3^*J*_C-F_ = 5.2, C-5), 134.12, 134.00, 133.75, 129.98, 129.77, 128.83, 128.76, 128.71 (Ph), 87.22 (C-1′), 81.39 (C-4′), 74.19 (C-3′), 71.49 (C-2′), 63.48 (C-5′). HRMS: *m*/*z* [C_31_H_22_ClFN_4_O_7_+H^+^] calculated 617.1234, found 617.1237, [C_13_H_22_ClFN_4_O_7_+Na^+^] calculated 639.1053, found 639.1054, [C_13_H_22_ClFN_4_O_7_+K^+^] calculated 655.0793, found 655.0792.



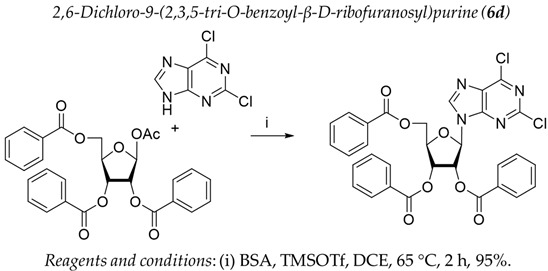



For synthesis of compound **6d** to a suspension of 0.390 g (2.063 mmol) of 2,6-dichloropurine in 20 mL of DCE, 1.039 g (2.058 mmol) of 1-*O*-acetyl-2,3,5-tri-*O*-benzoyl-β-d-ribofuranose and *N*,*O*-bis-trimethylsilylacetamide (BSA, 0.610 mL, 2.495 mmol) were added. The resulting mixture was stirred for approximately 30 min at r.t. until homogenization and trimethylsilyltriflate (TMSOTf, 0.51 mL, 2.811 mmol) was then added to the mixture and the mixture was heated at 65 °C in an oil bath for 2 h. The resulting mixture was cooled to r.t., diluted with ethyl acetate, washed successively with a saturated sodium bicarbonate solution (50 mL), water (50 mL), brine (50 mL), dried over sodium sulfate, filtered and evaporated in a vacuum to dryness. The resulting residue (yellowish foam) was purified chromatographically on silica gel in dichloromethane/ dichloromethane-ethanol 100 −100:5 (*v*:*v*) to obtain 1,238 g (95%) of 2,6-dichloro-9-(2,3,5-tri-*O*-benzoyl-β-d-ribofuranosyl)purine as a foam. ^1^H-NMR (300 MHz, DMSO-*d*_6_): δ = 8.28 s (1H, H-8), 8.06 dd (2H, *^3^J* = 8.3, *^4^J* = 1.3, ortho-H-Bz), 8.03 dd (2H, *^3^J* = 8.3, *^4^J* = 1.2, ortho-H-Bz), 7.93 dd (2H, *^3^J* = 8.3, *^4^J* = 1.3 ortho-H-Bz), 7.64–7.53 m (3H, para-H-Bz), 7.49–7.41 m (4H, meta-H-Bz), 7.38 dd (2H, *^3^J*_1_ = 8.4, *^3^J*_1_ = 7.9, meta-H-Bz), 6.48 d (1H, *J*_1′2′_ = 5.4, H-1′), 6.18 dd (1H, *J*_2′1′_ = 5.4, *J*_2′3′_ = 5.8, H-2′), 6.13 dd (1H, *J*_3′2′_ = 5.8, *J*_3′4′_ = 4.2, H-3′), 4.92 dd (1H, *J*_5a′5′b_ = −12.2, *J*_5′a4′_ = 3.3, H-5′a), 4.87 ddd (1H, *J*_4′3′_ = 4.2, *J*_4′5′a_ = 3.3, *J*_4′5′b_ = 3.3, H-4′), 4.73 dd (1H, *J*_5a′5′b_ = −12.2, *J*_5′a4′_ = 3.3, H-5′a). ^13^C NMR (75 MHz, CDCl_3_): 166.20 (C=O, 5′*O*-Bz), 165.45, 165.29 (C=O, Bz), 153.65 (C-2), 152.78 (C-4), 152.50 (C-6), 143.96 (C-8), 131.07 (C-5), 134.14, 134.03, 133.78, 130.06, 130.02, 129.78, 128.88, 128.78, 128.75 (Ph), 87.17 (C-1′), 81.69 (C-4′), 74.47 (C-3′), 71.72 (C-2′), 63.77 (C-5′). HRMS: *m*/*z* [C_31_H_22_Cl_2_N_4_O_7_+H^+^] calculated 633.0938, found 633.0934, [C_13_H_22_Cl_2_N_4_O_7_+Na^+^] calculated 655.0758, found 655.0753, [C_13_H_22_Cl_2_N_4_O_7_+K^+^] calculated 671.0497, found 671.0492.

### 3.2. Biology

#### 3.2.1. Real-Time Detection of Tdp1 Activity

Purified human recombinant Tdp1 was prepared as described earlier [64]. Real-time detection of Tdp1 activity was reported in our previous work [43]. The approach consists of a fluorescence intensity measurement of the reaction of quencher removal from the fluorophore-quencher coupled DNA oligonucleotide catalyzed by Tdp1 in the presence of the inhibitor (the control samples contained 1% DMSO). The biosensor was single-stranded oligonucleotide (5′-[FAM] AAC GTC AGGGTC TTC C [BHQ]-3′) containing fluorophore at the 5′ end (5,6-FAM) and a Black Hole Quencher 1 (BHQ) at the 3′ end, and was synthesized in the Laboratory of nucleic acid chemistry at the Institute of Chemical Biology and Fundamental Medicine SB RAS, Novosibirsk, Russia. The reactions were incubated at a constant temperature 25 °C in a POLARstar OPTIMA fluorimeter, BMG LABTECH, GmbH, to measure fluorescence every 55 s (Ex485/Em520 nm) during the linear phase (here, the data are from 0 to 8 min). To determine the values of half maximal inhibitory concentration (IC_50_) reaction mixtures (200 μL) contained Tdp1 buffer (50 mM Tris-HCl, pH 8.0, 50 mM NaCl, 7 mM β-mercaptoethanol), 50 nM oligonucleotide, testable inhibitor and purified Tdp1 (1.5 nM). The IC_50_ values were determined using a six-point concentration response curve in three independent experiments and calculated using MARS Data Analysis 2.0 (BMG LABTECH).

To determine the kinetic parameters: apparent maximum rate of enzymatic reaction (V_max_), Michaelis constant (K_M_), and possible inhibition mechanism, steady-state kinetic experiments of Tdp1 enzymatic reaction were carried out at five fixed concentrations of the substrate with various inhibitor concentrations. The standard reaction mixtures (200 μL) contained 56 nM, 96 nM, 163 nM, 277 nM, 470 nM or 800 nM oligonucleotide substrate, testable inhibitor, 1.5 nM recombinant human Tdp1 and reaction buffer components (50 mM Tris-HCl, pH 8.0, 50 mM NaCl, 7 mM β-mercaptoethanol). After the addition of the enzyme, the reaction mixtures were incubated at a constant temperature of 26 °C and measured in a POLARstar OPTIMA fluorimeter, BMG LABTECH, GmbH, to measure fluorescence every 55 s (Ex485/Em520 nm) during the linear phase (data from 0 to 8 minutes). The initial data (kinetic curves) were obtained in three independent experiments and statistically processed in the software OriginPro 8.6.0.

#### 3.2.2. Investigation of PARP1/2 Activity by Inclusion of α-[^32^P]-NAD^+^ in Poly(ADP-ribose)

Purified human recombinant PARP1/2 [65] and [^32^P] NAD^+^ [27] were prepared as described earlier. The activity of PARP1/2 was evaluated by the rate of incorporation of the radioactive label into the acid-insoluble reaction product—PAR as described previously [27]. 

#### 3.2.3. Alkaline Comet Assay

HeLa cell culture was grown in a 24-well plate to a concentration of 0.05 million/mL, then treated with the tested compounds and incubated for 1 h. To obtain a cell suspension, the cells were treated with trypsin (30 µL in each well for 5 min at 37 °C), and then 200 µL of PBS + 10% FBS were added to each well.

The resulting cell suspension (50 µL) was mixed with 250 µL of a 1% solution of molten low melting agarose (CertifiedTM LMAgarose; BIO-RAD), 250 µL of this mixture was transferred to the slides pre-coated with a 1% normal melting agarose (Agarose; Helicon), and left for several minutes at 4 °C to solidify. The resulting slides were incubated in a lysing solution (2.5 M NaCl, 100 mM EDTA, 10 mM Tris base, 1% Triton, 5% DMSO, pH 10.0) for an hour at 4 °C. Then the slides were immersed in an electrophoretic buffer (300 mM NaOH, 1 mM EDTA, pH > 13) for 45 min at 4 °C. Electrophoresis was performed at 20 V, 450 mA for 10 min while cooling on ice. After that, the slides were immersed 2 times for 5 min in 0.4 M Tris-HCl pH 7.5 solution for neutralization, washed twice with cold water, and stained with SYBR Green I (excitation maximum at 497 nm, emission maximum at 520 nm; Thermo Fisher Scientific KK).

The images were obtained using a CELENA © S digital microscope (Logos Biosystems, Inc., Annandale, VA, USA) and analyzed with Comet analysis software (Trevigen, Inc., Gaithersburg, MD, USA. In all experiments, 2 slides were evaluated for each sample by counting at least 200 cells on the glass. As a parameter for measuring the level of DNA damage, the median value of the percentage of DNA in the tail was chosen (% Tail DNA = 100 × the intensity of DNA fluorescence in the tail of the comet/the intensity of fluorescence of the total DNA of the comet).

#### 3.2.4. Cell Culture Cytotoxicity Assay

Cytotoxicity of the compounds was examined on human cell lines HeLa using an EZ4U colorimetric test (Biomedica, Vienna, Austria). The HeLa cell line was obtained from the Russian Cell Culture Collection (RCCC) Institute of Cytology RAS, St. Petersburg, Russia. The cells were grown in DMEM/F12 medium (Gibco, Thermo Fisher Scientific, Waltham, MA, USA), with 1x GlutaMAX (Gibco, Thermo Fisher Scientific, Waltham, MA, USA), 50 IU/mL penicillin, 50 μg/mL streptomycin (MP Biomedicals), and in the presence of 10% fetal bovine serum (Biolot, Saint-Petersburg, Russia) in a 5% CO_2_ atmosphere. Since lipophilic nucleosides are soluble in DMSO, cells were grown in the presence of 1% DMSO in the control wells. After the formation of a 30–50% monolayer, the tested compounds were added to the medium, and the cell culture was monitored for 3 days. The values were normalized to their own control in each case. At least three independent experiments were carried out. The 50% cytotoxic concentration (CC_50_) was defined as the compound concentration that reduced the cell viability by 50% when compared to the untreated controls. The CC_50_ value was determined using OriginPro 8.6.0 software (OriginLab, Northampton, MA, USA). The measurements were carried out in three independent experiments.

### 3.3. Experiments In Vivo

#### 3.3.1. Lab Animals

Three- to four-month-old male and female C57Bl/6 mice (19–21 g) from the breeding colony of the Institute of Cytology and Genetics, SB RAS, Novosibirsk, Russia, were used in the study. The animals were kept on sawdust in plastic cages of 5–7 mice per cage with free access to ground food (“Laboratorkorm”, Moscow, Russia) and tap water. All procedures with the mice were performed in accordance with the international rules for experiments on animals [European Communities Directive (86/609 EEC)].

#### 3.3.2. Tumor Models

The experimental tumor model was transplantable Krebs-2 carcinoma obtained from the cell depository of the Institute of Cytology and Genetics SB RAS (Novosibirsk, Russia) and was maintained in mice as a transplanted tumor. Krebs-2 was obtained from epithelial cells of the abdominal wall of a mouse. Krebs-2 carcinoma is nonspecific in its host requirements and can be grafted into mice with any genetic context. With intraperitoneal inoculation, it forms an ascites form. The tumor is weakly immunogenic for mice of all strains, does not metastasize [55,56,66,67].

Prior to the transplantation, the tumor was minced by means of scissors and passed through a sieve with a diameter 0.914 mm. The tumor tissue was resuspended in 0.9% NaCl solution and injected intraperitoneally in a volume of 0.1 mL. 

The antitumor effect was assessed by the weight of the ascites and by the number of tumor cells in ascites. The weight of Krebs-2 ascites was assessed by the difference in the weight of mice before and after the removal of ascites. The number of tumor cells in ascites was determined using a hemocytometer Goryaev camera. In autopsied mice, the weights of liver, lung, heart, kidneys, and spleen were assessed as well as the tumor and body weights.

#### 3.3.3. Treatment

The animals were treated once in total dose of Tpc 1 mg/kg and Tdp1 inhibitor (**5f**, **6c** or **6d**) 100 mg/kg. We started the treatment on the fourth day after the tumor was inoculated, with the expectation that it had just begun to form. Both Tpc and the compound were administered intraperitoneally (i/p). Tdp1 inhibitors were given in a DMSO–Tween 80 suspension (0.2 mL/mouse). The animals were euthanized eight days after the transplantation and the compound effect on mice was evaluated by measuring the tumor weight and number of tumor cells in ascites.

## 4. Conclusions

In the present work, we have shown for the first time that a derivative of a nucleoside inhibiting Tdp1 sensitizes mice Krebs-2 ascitic carcinoma cells to the action of the Top1 inhibitor Tpc used in the clinic. The leader compound **6d** does not cause DNA damage by itself, but enhances the DNA-damaging ability of Tpc, allowing the dose of the latter to be reduced by a factor of 10 to achieve the same effect. It has also been shown that this compound inhibits Tdp1 by an uncompetitive mechanism, that is, it binds to the enzyme-substrate complex, which suggests a high specificity of this inhibitor compared with competitive inhibitors. Thus, this type of compound is a promising platform for further development of drugs for the accompanying cancer therapy in combination with Top1 inhibitors.

## Figures and Tables

**Figure 1 molecules-28-00323-f001:**
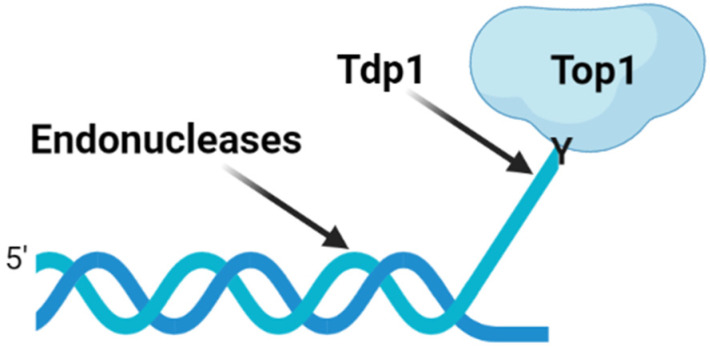
Pathways for the removal of irreversible topoisomerase cleavage complexes. After proteolysis and/or denaturation of Top1, Tdp1 can hydrolyze the tyrosyl-DNA links. In addition, the endonucleases can remove the DNA segment covalently attached to topoisomerases. Created with Biorender.

**Figure 2 molecules-28-00323-f002:**
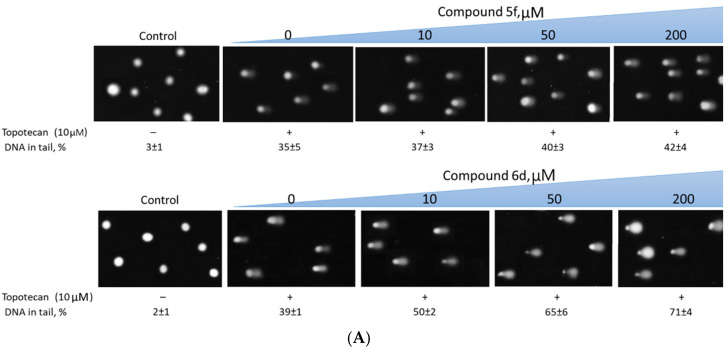
The influence of compounds **5f** and **6d** on the accumulation of DNA damage induced by topotecan. (**A**) The images of cells treated with combination of topotecan and leader compounds in different concentrations. (**B**) The dependence of tail DNA on Tdp1 inhibitor concentration. The dose-dependent increase of % tail DNA with 6d concentration can be observed. Error bars show SD for two independent experiments.

**Figure 3 molecules-28-00323-f003:**
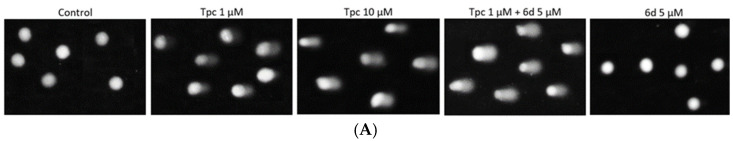
The use of compound **6d** in a non-toxic concentration reduces the dose of topotecan required to produce a serious level of DNA damage by 10 times. The compound **6d** leads to similar DNA damage accumulation in cases of treatment with only Tpc 10 μM and a combination of Tpc 1 μM and **6d** 5 μM (pictures (**A**) and histogram (**B**)), * *p* < 0.05. Since cells treated with DMSO did not differ from the untreated control, we decided not to present them here in order not to overload the picture. There is a photo of cells treated with DMSO in Appendix A.

**Figure 4 molecules-28-00323-f004:**
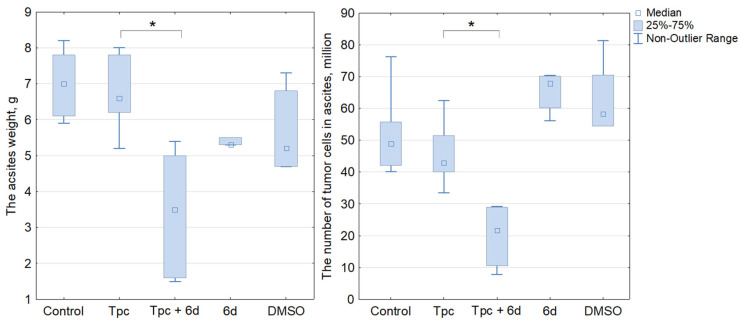
Investigation of the influence of compound **6d** on the antitumor effect of topotecan in vivo on a Krebs-2 carcinoma model. On the left are data on the weight of ascites; on the right, the number of tumor cells in ascites, * *p* < 0.05. Compound **6d** in combination with Tpc led to a synergistic decrease of tumor size compared to only Tpc, while the Tdp1 inhibitor itself did not affect tumor growth.

**Table 1 molecules-28-00323-t001:** Inhibitory activities of compounds against Tdp1 and their cytotoxicity against HeLa cells. The half maximal inhibitory concentration (IC50) is a measure of the potency of a substance in inhibiting. IC50 is a quantitative measure that indicates how much of an inhibitor is needed to inhibit enzymatic activity by 50%. Moreover, 50% cytotoxicity concentration (CC50) is the concentration of test compounds required to reduce cell viability by 50%.

No	Structure	Tdp1 IC_50_, μM	CC_50_, μM	Enhancement of Tpc Cytotoxicity by 10 μM Compounds, Times
Natural compounds				
**1**	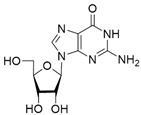	>50	nd *	nd *
**2**	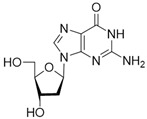	>50	nd	nd
**3**	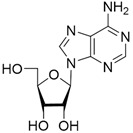	>50	nd	nd
**4**	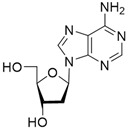	>50	nd	nd
Semisynthetic compounds				
**5a**	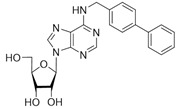	>50	>100	2.5
**5b**	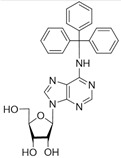	>50	59.6 ± 0.6	1.7
**5c**	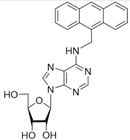	22 ± 3	58 ± 2	2.2
**5d**	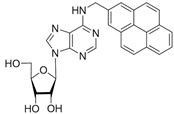	>50	56 ± 10	3.1
**5f**	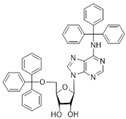	0.30 ± 0.06	>100	1.7
**6a**	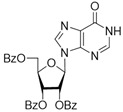 **	7 ± 1	58 ± 5	1.8
**6b**	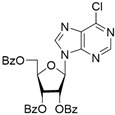	2.0 ± 0.6	56 ± 26	1.9
**6c**	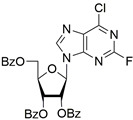	1.0 ± 0.2	35 ± 20	6.5
**6d**	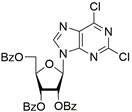	0.82 ± 0.02	33 ± 15	4.8

* nd—not determined; ** OBz—benzoyl group

## Data Availability

The data presented in this study are available upon request from the corresponding author.

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
