# Peer review of "The Lipophilic Purine Nucleoside—Tdp1 Inhibitor—Enhances DNA Damage Induced by Topotecan In Vitro and Potentiates the Antitumor Effect of Topotecan In Vivo"

_molecules, 2022, doi:10.3390/molecules28010323_

Round 1
Reviewer 1 Report
The manuscript by Chernyshova et al. describes the synthesis of lipophilic derivatives of purine nucleosides and their inhibitory effect on the DNA repair enzyme Tdp1, which corrects covalent topoisomerase 1(Top1)-DNA complexes. Based on this activity, some of the compounds are claimed to enhance the DNA damage inflicted by the Top1 inhibitor topotecan. Thus, the lead compounds might be useful as chemosensitizing drugs to be used in combination with the anti-cancer agent topotecan.
I must warn that I do not feel qualified to judge on the chemical synthesis and characterization of the compounds. The study seems generally correct, but suffers from some confusing descriptions and somehow carelessly presentation of the data, for instance Table 1 should include more explanations.
Abstract: “We suppose the Tdp1 inhibition would sensitize the effect of Top1 inhibitors” I find this sentence confusing, I would suggest: “We hypothesize that Tdp1 inhibition would sensitize cells towards the effect of...”
Introduction: Lane 37: “gene cytoreductive therapy” Perhaps this should be termed simply “cytoreductive therapy”, I do not understand the presence of the term “gene” there.
Fig. 1 legend: “... tyrosyl-DNA links following proteolysis and/or denaturation of Top1 or the endonucleases remove...” Confusing; the sentence should be split in two; something like “tyrosyl-DNA links. This is then followed by ...”
Lane 90: better in plural “... stabilizes PARP-DNA complexes”
Lane 96: what do you mean by “fluorescent screening”? This should be omitted in my opinion.
Section 2.1. Lane 106: Which compounds do you mean that “contain carbohydrate fragments at position 6 of adenine”? I cannot see them in Table 1. I do not know whether “Mitsunobu conditions” are so well known in the field; this sentence / part (referring to something previously done) needs a reference, anyway.
Lane 109: do you mean “derivatives 6a-6d”?
Lane 114: instead of “see Supplementary data” such supplementary figures should be specifically cited for every compound (e.g. here “S3A” and so on).
Table 1: I think that the table should be more fully explained, maybe by means of the legend: Define the terms and abbreviations used: IC50, CC50, BzO. The first 4 compounds are the natural nucleosides G dG, A, dA. This should be stated. Why are there two water molecules indicated in the first compound? It looks confusing that the ribose ring of the two last compounds has a different format.
Section 2.2. Lane 126: “... 13 aromatic and lipophilic derivatives...” I think that the first 4 compounds, being natural nucleosides, shouldn´t be considered “derivatives”...
Lane 128: a reference for the HBQ1 approach should be provided.
Section 2.3. Lane 153: Instead of “supposed” “intended”?
Lane 154: “...do not DISPLAY toxicity”
Lane 157: Use of test “EZ4U”: Instead of the commercial name, give a more general name or else the rationale / basis of this assay, and give a reference.
Lane 161: define CC50 the first time this term is used (in Table 1 and/or here)
Lane 176: “... in the second and SIXTH positions”?
Lane 177: “... we took THE three most potent...”
Section 2.4. Lane 227: Better: “... SCAN1 develops for decades” or “takes decades to develop”
Section 2.5: I find lanes 232-241 of the first paragraph confusing and contradictory, this part should be rewritten: First you say that you perform DNA damage assays with the three “lead compounds” (by the way, they have not been defined as “lead” in the previous sections) and then you say that you do not do it with 6c. The first sentence is not well built, I think. It could say something like “In the next step, we performed (“simultaneously can be omitted”) in vitro assays on... and in vivo experiments on the influence of lead compounds” (molecules X,Y,Z)...” “Since we did not OBTAIN sensitizing effect...”?
Lane 242: “At the moment” could be omitted.
Lane 246: “It is known that genotoxic agents...” A reference should be provided.
Lane 254: Instead of “...processes of formation of DNA damage...” GENERATION / INFLICTION...?
Lane 258: “amount OF DAMAGE”
Lane 259: Instead of “... began to recover these damages...” “... began to CORRECT these damages /LESIONS”?
Lanes 262-264: “... is ENOUGH to BE detected, reproduced and can be EVALUATED by ..the Comet assay AS LONG AS...THERE IS NO cell death...”?
Lane 271: “did not SIGNIFICANTLY differ..” (there is a small difference).
Fig. 2 legend: “... cells treated with A combination..” The last sentence is confusing: “A dose dependent increase ... of...WITH 6d concentration CAN BE OBSERVED ..”? Calculation of error bars should be stated.
Lane 286: Instead of “... the change of the damaged DNA amount” “the DEGREE of DNA damage...?
Section 2.6. Lane 321: instead of “inhibitor Tdp1” “Tdp inhibitor”. The same for Fig. 4 legend, which is very much a copy of this sentence.
I find the last paragraph (lanes 329-348) quite repetitive; I think it should be rewritten and shortened. Lane 329 “In our works we use the approach of developing..” This might be omitted: “We have developed...”
Materials and Methods. Section 3.1. Why a section 3.1.1 when there is no further one in 3.1?
Why are there several parts of the text in blue?
As mentioned previously, the specific supplementary graphs should be cited wherever they fit: data for compound 5f (S1) when talking about that compound and so on.
Lane 361 “... voltage e”?
Lane 375. This part lacks some kind of introductory sentences for the compounds family 6, something like “For synthesis of compound 6b...”
I am not expert in organic chemistry, but I wonder whether the reactions should be cited with different numbers.. In SOCl2, “2” should be subindex.
Lane 392: I think that there is an inconsistency with Fig. S3C.
Section 3.2. I do not understand the difference between the two modes of Tdp1 enzymatic assays. This should be better explained in the Results section. See below about Fig. S6.
Supplementary information: As mentioned, the figure panels need more explanations in the labels. Fig. S3B: I guess this is a 13C-NMR spectrum, not 1H. The MS graphs need more explanation: I do not understand the meaning of further main peaks apart from the mentioned ones (Na+ and K+ adducts). Fig. S6: I do not see that the left graphs show Tdp1 reaction rates. The S concentration points do not match those stated under M&M. Fig. S9 “itself”?
Author Response
Dear Editors and Reviewers,
Thank you very much for the consideration of our manuscript "The lipophilic purine nucleosides – Tdp1 inhibitors – enhance DNA damage induced by topotecan in vitro and potentiate the antitumor effect of topotecan in vivo" (Manuscript ID: molecules-2030489, Authors: Irina A. Chernyshova, et al.) for publication in Molecules (MDPI). We have considered all the Reviewers’ comments very carefully, and we provide a revised version of the manuscript for further consideration. We are grateful to all three Reviewers for their comments, all of them were all taken into account. All Reviewers, especially Reviewer 1, made a great job to improve the quality of our manuscript to make it more understandable for other researchers. The revised version of the manuscript is an improvement over the original, and we hope that the improved version would be worth publishing in Molecules. Our detailed point-by-point answers to the Reviewer’s comments are as follows.
Sincerely,
Olga Lavrik and coauthors
Review 1
The manuscript by Chernyshova et al. describes the synthesis of lipophilic derivatives of purine nucleosides and their inhibitory effect on the DNA repair enzyme Tdp1, which corrects covalent topoisomerase 1(Top1)-DNA complexes. Based on this activity, some of the compounds are claimed to enhance the DNA damage inflicted by the Top1 inhibitor topotecan. Thus, the lead compounds might be useful as chemosensitizing drugs to be used in combination with the anti-cancer agent topotecan.
I must warn that I do not feel qualified to judge on the chemical synthesis and characterization of the compounds. The study seems generally correct, but suffers from some confusing descriptions and somehow carelessly presentation of the data, for instance Table 1 should include more explanations.
Abstract: “We suppose the Tdp1 inhibition would sensitize the effect of Top1 inhibitors” I find this sentence confusing, I would suggest: “We hypothesize that Tdp1 inhibition would sensitize cells towards the effect of...”
- Thank you! It was done.
Introduction: Lane 37: “gene cytoreductive therapy” Perhaps this should be termed simply “cytoreductive therapy”, I do not understand the presence of the term “gene” there.
Fig. 1 legend: “... tyrosyl-DNA links following proteolysis and/or denaturation of Top1 or the endonucleases remove...” Confusing; the sentence should be split in two; something like “tyrosyl-DNA links. This is then followed by ...”
- It was done.
Lane 90: better in plural “... stabilizes PARP-DNA complexes”
- It was done.
Lane 96: what do you mean by “fluorescent screening”? This should be omitted in my opinion.
Section 2.1. Lane 106: Which compounds do you mean that “contain carbohydrate fragments at position 6 of adenine”? I cannot see them in Table 1. I do not know whether “Mitsunobu conditions” are so well known in the field; this sentence / part (referring to something previously done) needs a reference, anyway.
- Of course, not a carbohydrate, but a hydrocarbon! We are sorry, it was corrected.
Thank you for this comment. Of course hydrocarbon, not carbohydrate. This is mistake, we corrected it. Therefore, we added missed numbers for previously “carbohydrate” and now “hydrocarbon” 5c-d in the corresponding part of the text for clarity. Compounds 5c-d were previously described in our work (see Ref.Orlov et al. Bioorg Med Chem Lett 2017). As compound 5a was not described but was obtained according to the known literature methods we provided an additional experimental procedure for its synthesis and spectral data in Supplementary as well. We also added an additional reference on synthesis of tetraacetyladenosine substrate for alkylation (see Tararov et al. Synthesis 2011).
Lane 109: do you mean “derivatives 6a-6d”?
Lane 114: instead of “see Supplementary data” such supplementary figures should be specifically cited for every compound (e.g. here “S3A” and so on).
- It was done.
Table 1: I think that the table should be more fully explained, maybe by means of the legend: Define the terms and abbreviations used: IC50, CC50, BzO. The first 4 compounds are the natural nucleosides G dG, A, dA. This should be stated. Why are there two water molecules indicated in the first compound? It looks confusing that the ribose ring of the two last compounds has a different format.
- Thank you, we made the changes. Hope now this table is more understandable. We have deciphered all the designations in the header of the table or in the note below it. Natural and semi-synthetic compounds now are indicated directly in the table. Water molecules from compound 1 were removed. We unified the format of our compounds 6c and 6d with other structures.
Section 2.2. Lane 126: “... 13 aromatic and lipophilic derivatives...” I think that the first 4 compounds, being natural nucleosides, shouldn´t be considered “derivatives”...
- Thank you! It was corrected.
Lane 128: a reference for the HBQ1 approach should be provided.
- It was done.
Section 2.3. Lane 153: Instead of “supposed” “intended”?
Lane 154: “...do not DISPLAY toxicity”
Lane 157: Use of test “EZ4U”: Instead of the commercial name, give a more general name or else the rationale / basis of this assay, and give a reference.
- It was done.
Lane 161: define CC50 the first time this term is used (in Table 1 and/or here)
- CC50 was deciphered in the table header.
Lane 176: “... in the second and SIXTH positions”?
Lane 177: “... we took THE three most potent...”
Section 2.4. Lane 227: Better: “... SCAN1 develops for decades” or “takes decades to develop”
Section 2.5: I find lanes 232-241 of the first paragraph confusing and contradictory, this part should be rewritten: First you say that you perform DNA damage assays with the three “lead compounds” (by the way, they have not been defined as “lead” in the previous sections) and then you say that you do not do it with 6c. The first sentence is not well built, I think. It could say something like “In the next step, we performed (“simultaneously can be omitted”) in vitro assays on... and in vivo experiments on the influence of lead compounds” (molecules X,Y,Z)...” “Since we did not OBTAIN sensitizing effect...”?
- Thank you! We have rewritten the beginning of section 2.5. We hope it looks more logical now.
Lane 242: “At the moment” could be omitted.
Lane 246: “It is known that genotoxic agents...” A reference should be provided.
- We thank Reviewer 1 for the vigilance. The phrase "It is known that genotoxic agents..." is not entirely accurate, so we removed it and rewrote the sentence.
Lane 254: Instead of “...processes of formation of DNA damage...” GENERATION / INFLICTION...?
Lane 258: “amount OF DAMAGE”
Lane 259: Instead of “... began to recover these damages...” “... began to CORRECT these damages /LESIONS”?
Lanes 262-264: “... is ENOUGH to BE detected, reproduced and can be EVALUATED by ..the Comet assay AS LONG AS...THERE IS NO cell death...”?
- Thank you! We have rewritten this sentence. It seems to us that it has become clearer.
Lane 271: “did not SIGNIFICANTLY differ..” (there is a small difference).
- Corrected.
Fig. 2 legend: “... cells treated with A combination..” The last sentence is confusing: “A dose dependent increase ... of...WITH 6d concentration CAN BE OBSERVED ..”? Calculation of error bars should be stated.
Lane 286: Instead of “... the change of the damaged DNA amount” “the DEGREE of DNA damage...?
Section 2.6. Lane 321: instead of “inhibitor Tdp1” “Tdp inhibitor”. The same for Fig. 4 legend, which is very much a copy of this sentence.
I find the last paragraph (lanes 329-348) quite repetitive; I think it should be rewritten and shortened. Lane 329 “In our works we use the approach of developing..” This might be omitted: “We have developed...”
- We have rewritten the paragraph and we believe it has become more readable and logical.
Materials and Methods. Section 3.1. Why a section 3.1.1 when there is no further one in 3.1?
Why are there several parts of the text in blue?
As mentioned previously, the specific supplementary graphs should be cited wherever they fit: data for compound 5f (S1) when talking about that compound and so on.
- We replaced "see supplementary data" with direct links to images in supplementary.
Lane 361 “... voltage e”?
- Corrected to voltage.
Lane 375. This part lacks some kind of introductory sentences for the compounds family 6, something like “For synthesis of compound 6b...”
- It was done.
I am not expert in organic chemistry, but I wonder whether the reactions should be cited with different numbers. In SOCl2, “2” should be subindex.
- We added the schemes with reactions in M&M part and corrected subindex.
Lane 392: I think that there is an inconsistency with Fig. S3C.
- Yes, you are right. We apologize, that full assessment was not provided as analysis of H+ and Na+ ionic modes was given on a separate list. Therefore, we added this new information to Fig. S3C (novel number S4C because of addition of spectral data for 5a). It fully corresponds to the HRMS data described in M&M section. We also added the corresponding ionic modes for other HRMS, where necessary (see Supplementary Data).
Section 3.2. I do not understand the difference between the two modes of Tdp1 enzymatic assays. This should be better explained in the Results section. See below about Fig. S6.
- We tried to make it more clear in M&M section.
Supplementary information:
As mentioned, the figure panels need more explanations in the labels.
- We deciphered KM and Vmax.
Fig. S3B: I guess this is a 13C-NMR spectrum, not 1H.
- Corrected.
The MS graphs need more explanation: I do not understand the meaning of further main peaks apart from the mentioned ones (Na+ and K+ adducts).
- We added determination for peaks in Supplementary as commentaries below each picture.
HRMS spectra of pure stable compounds usually represent one or three peaks corresponding to H+, Na+ and K+ adducts (see newly added HRMS for biphenyladenosine 5a, Fig. S1C). Registration of HRMS spectra of fluorine and chlorine containing compounds is accompanied with difficulties associated with their reactivity due to the presence of fluorine and chlorine atoms. Compounds 6b-d are stable for several weeks in solutions at ambient temperature according to NMR-spectra, but they can interact with solvents by Сl/F substitution or elimination at elevated temperature (gas-liquid chromatography under heated nitrogen flow at 180°C undoubtly favours these processes, see General Part of Materials and Methods). There are many possibilities of molecular destruction under mass-spectrometry analysis, associated with elimination of electron-withdrawal groups and molecular fragments in complex with solvent, formed on previous on LC chromatographic step.
In our HRMS-spectra 6b-d we observed several additional peaks. Peaks corresponding to 663 and 680 are difficult to precisely interpret due to possibilities of formation of several adducts, but they all indicate the presence of solvent molecule (acetonitrile and water) in a complex. There is also a possibility to form double-charged and thriple-charged molecular ions due to the presence of 4 nitrogen atoms in nucleosides (m/z 644). Peaks with m/z 628, 614 and 608 are derived from 663 by elimination of chlorine or deamination or purine base destruction with an accuracy of one thousandth which can be also evidence for halogen-substituted derivatives.
Fig. S6: I do not see that the left graphs show Tdp1 reaction rates.
- The left column of the graphs shows the dependence of the reaction rate (in a linear range) on the concentration of the substrate in the presence of specific concentrations of the inhibitor. We corrected y-
The S concentration points do not match those stated under M&M.
- Thank you for notice it! We corrected concentrations in M&M part.
Fig. S9 “itself”?
- We removed it and rewrote the sentence.
Reviewer 2 Report
The paper provides important informations about the lipophilic purine nucleosides – Tdp1 inhibitors – enhance 2 DNA damage induced by topotecan in vitro and potentiate the 3 antitumor effect of topotecan in vivo. The structure of the article is well collated, following a logical and easy to interpret. In addition, tables and figures are clear, easy interpret and allows the reader to quickly locate what they are trying to determine. However, there are a few comment I would like to make for further clarification and scrutiny.
Lines 2-4; in vitro in vivo must be written as italic.
Line 99; The aim of the study must be written more clearly.
Line 119; Please give the reference number for Vorbrüggen and co-workers.
Author Response
Dear Editors and Reviewers,
Thank you very much for the consideration of our manuscript "The lipophilic purine nucleosides – Tdp1 inhibitors – enhance DNA damage induced by topotecan in vitro and potentiate the antitumor effect of topotecan in vivo" (Manuscript ID: molecules-2030489, Authors: Irina A. Chernyshova, et al.) for publication in Molecules (MDPI). We have considered all the Reviewers’ comments very carefully, and we provide a revised version of the manuscript for further consideration. We are grateful to all three Reviewers for their comments, all of them were all taken into account. All Reviewers made a great job to improve the quality of our manuscript to make it more understandable for other researchers. The revised version of the manuscript is an improvement over the original, and we hope that the improved version would be worth publishing in Molecules. Our detailed point-by-point answers to the Reviewer’s comments are as follows.
Sincerely,
Olga Lavrik and coauthors
Review 2
The paper provides important informations about the lipophilic purine nucleosides – Tdp1 inhibitors – enhance 2 DNA damage induced by topotecan in vitro and potentiate the 3 antitumor effect of topotecan in vivo. The structure of the article is well collated, following a logical and easy to interpret. In addition, tables and figures are clear, easy interpret and allows the reader to quickly locate what they are trying to determine. However, there are a few comment I would like to make for further clarification and scrutiny.
Lines 2-4; in vitro in vivo must be written as italic.
Line 99; The aim of the study must be written more clearly.
- It was d
Line 119; Please give the reference number for Vorbrüggen and co-workers.
- Thank you for notice it! We added it in references list.
Reviewer 3 Report
The manuscript by Chernyshova I. et al. “The lipophilic purine nucleosides – Tdp1 inhibitors – enhance DNA damage induced by topotecan in vitro and potentiate the antitumor effect of topotecan in vivo” contains an interesting study in the field of the development of anticancer drug cocktails aimed at DNA repair enzymes.
The experiments are well done. Authors started from the synthesis of a set of new lipophilic nucleoside compounds and the stepwise selection of a lead compounds. Finally, a good sensitizer of topotecan was found both in vitro and in vivo among the set of synthesized compounds. The most active Tdp1 inhibitor compound 6d enhanced the antitumor activity of topotecan. The undoubted advantage of the work is the discovery of the compound 6d that can reduce the dose of a sufficiently toxic topotecan to achieve the same effect. In my opinion the manuscript can be accepted after minor revision.
Minor comment
- In the title it is better not to write “Tdp1 inhibitors” in plural, because only one the best compound was discovered as topotecan enhancer in vitro and in vivo.
- In the description of various experiments, it is indicated that DMSO was used in the control. This suggests that TDP1 inhibitors are soluble in DMSO. First, it would be nice to write about it explicitly so that the reader does not have to guess. Secondly, the description of the comets does not indicate whether the control cells were treated with DMSO, and if so, at what concentration. Please, indicate it.
Author Response
Dear Editors and Reviewers,
Thank you very much for the consideration of our manuscript "The lipophilic purine nucleosides – Tdp1 inhibitors – enhance DNA damage induced by topotecan in vitro and potentiate the antitumor effect of topotecan in vivo" (Manuscript ID: molecules-2030489, Authors: Irina A. Chernyshova, et al.) for publication in Molecules (MDPI). We have considered all the Reviewers’ comments very carefully, and we provide a revised version of the manuscript for further consideration. We are grateful to all three Reviewers for their comments, all of them were all taken into account. All Reviewers made a great job to improve the quality of our manuscript to make it more understandable for other researchers. The revised version of the manuscript is an improvement over the original, and we hope that the improved version would be worth publishing in Molecules. Our detailed point-by-point answers to the Reviewer’s comments are as follows.
Sincerely,
Olga Lavrik and coauthors
Review 3
The manuscript by Chernyshova I. et al. “The lipophilic purine nucleosides – Tdp1 inhibitors – enhance DNA damage induced by topotecan in vitro and potentiate the antitumor effect of topotecan in vivo” contains an interesting study in the field of the development of anticancer drug cocktails aimed at DNA repair enzymes.
The experiments are well done. Authors started from the synthesis of a set of new lipophilic nucleoside compounds and the stepwise selection of a lead compounds. Finally, a good sensitizer of topotecan was found both in vitro and in vivo among the set of synthesized compounds. The most active Tdp1 inhibitor compound 6d enhanced the antitumor activity of topotecan. The undoubted advantage of the work is the discovery of the compound 6d that can reduce the dose of a sufficiently toxic topotecan to achieve the same effect. In my opinion the manuscript can be accepted after minor revision.
Minor comment
- In the title it is better not to write “Tdp1 inhibitors” in plural, because only one the best compound was discovered as topotecan enhancer in vitro and in vivo.
- The title was changed.
- In the description of various experiments, it is indicated that DMSO was used in the control. This suggests that TDP1 inhibitors are soluble in DMSO. First, it would be nice to write about it explicitly so that the reader does not have to guess. Secondly, the description of the comets does not indicate whether the control cells were treated with DMSO, and if so, at what concentration. Please, indicate it.
- We have indicated which controls used DMSO in both the result and M&M sections.